# 3D Segmenter: 3D Transformer based Semantic Segmentation via 2D Panoramic Distillation

**Zhennan Wu**[1,2]**, Yang Li**[1]**, Yifei Huang**[1,3]**, Lin Gu** [*2,1] **, Tatsuya Harada**[1,2]**, Hiroyuki Sato**[1]
[1] The University of Tokyo
[2] RIKEN Center for Advanced Intelligence Project
[3] Shanghai AI Laboratory

## Abstract

Recently, 2D semantic segmentation has witnessed a significant advancement thanks to the huge amount of 2D image datasets available. Therefore, in this work, we propose the first 2D-to-3D knowledge distillation strategy to enhance 3D semantic segmentation model with knowledge embedded in the latent space of powerful 2D models. Specifically, unlike standard knowledge distillation, where teacher and student models take the same data as input, we use 2D panoramas properly aligned with corresponding 3D rooms to train the teacher network and use the learned knowledge from 2D teacher to guide 3D student. To facilitate our research, we create a large-scale, fine-annotated 3D semantic segmentation benchmark, containing voxel-wise semantic labels and aligned panoramas of 5175 scenes. Based on this benchmark, we propose a 3D volumetric semantic segmentation network, which adapts Video Swin Transformer as backbone and introduces a skip connected linear decoder. Achieving a state-of-the-art performance, our 3D Segmenter is computationally efficient and only requires 3.8% of the parameters compared to the prior art. Our code is released on https://github.com/swwzn714/3DSegmenter.

## 1 Introduction

Semantic segmentation assigns each 2D pixel(Long et al., 2015) or 3D point(Qi et al., 2017a)/voxel(Çiçek et al., 2016) to a separate category label representing the corresponding object class. As a fundamental computer vision technique, semantic segmentation has been widely applied to medical image analysis(Ronneberger et al., 2015), autonomous driving(Cordts et al., 2016) and robotics(Ainetter & Fraundorfer, 2021). Most of the existing efforts are invested in 2D settings thanks to great amount of public 2D semantic segmentation datasets(Zhou et al., 2017; Cordts et al., 2016; Nathan Silberman & Fergus, 2012). Nowadays, the wide availability of consumer 3D sensors also largely promotes the need for 3D semantic segmentation (Han et al., 2020; Choy et al., 2019; Thomas et al., 2019; Graham et al., 2018b; Nekrasov et al., 2021).

We have seen great success in semantic segmentation in 2D images. However, such success does not fully transfer to the 3D domain. One reason is that, given the same scene, processing the 3D data usually requires orders of magnitude more computations than processing a 2D image. *E.g.* using a volumetric 3D representation, the number of voxels grows $\mathcal{O}(n^3)$ with the size of the scene. As a result, existing 3D semantic segmentation methods usually need to use smaller receptive fields or shallower networks than 2D models to handle large scenes.

This motivates us to facilitate 3D semantic segmentation using 2D models. Specifically, we propose a novel 2D-to-3D knowledge distillation method to enhance 3D semantic segmentation by leveraging the knowledge embedded in a 2D semantic segmentation network. Unlike traditional knowledge distillation approaches, where student and teacher models should take the same input, in our case, the 2D teacher model is pre-trained on a large-scale 2D image repository and finetuned by panoramas rendered from 3D scenes. The panorama image is rendered at the panorama of the 3D scans with

---
[*]Corresponding author

360° receptive field, i.e. it contains the global context of the environment. Our 3D student model takes the 3D scene as input and outputs 3D volumetric semantic segmentation. Through differentiable panoramic rendering of 3D semantic segmentation prediction, we could obtain the mapping from the 2D teacher's prediction to the 3D student's prediction. Then we transfer the class distributions of the pixel produced from the 2D teacher to the corresponding voxel of the 3D student. To the best of our knowledge, this is the first solution to distill from a pre-trained 2D model to enhance the 3D computer vision task.

To facilitate this 2D-to-3D knowledge distillation design, we create a large-scale indoor dataset called **PanoRooms3D**. PanoRooms3D contains 5917 rooms diversely furnished by 3D CAD objects and augmented by randomly sampled floor and wall textures. We manually filter out scenes that contain unlabeled furniture. We prepare for each room a dense 3D volume, and a corresponding 2D panorama that is rendered in the center of the room with 360° receptive field, both containing clean semantic labels. Upon finding panorama-to-volume correspondence, the knowledge distillation from 2D to 3D is then enabled.

As our 3D student backbone, we propose a novel efficient architecture, 3D Segmenter, for semantic segmentation of 3D volumetric data. Inspired by Video Swin Transformer(Liu et al., 2022a), 3D Segmenter employ a pure-transformer structure with an efficient linear decoder. We demonstrate that our 3D Segmenter achieves superior performance and only requires 3.8% of the parameters compared to the prior art that uses 3D convolutional backbones.

Our contributions are three-fold:

- We propose the first 2D-to-3D knowledge distillation method to utilize the data-abundant and pretrain-ready 2D semantic segmentation to improve 3D semantic segmentation.

- We propose PanoRooms3D, a large-scale 3D volumetric dataset with a clean voxel-wise semantic label and well-aligned corresponding 2D panoramic renderings with pixel-wise semantic labels.

- We propose a novel efficient pure-transformer-based network for the 3D semantic segmentation task. Our 3D Segmenter outperforms prior art that uses 3D convolutional backbone with only 3.8% of the parameters.

We proved through experiments that our baseline variants have already achieved SoTA performance. The distilled knowledge from 2D further widened our lead. Considering the difficulty of 3D data collection, and the cubic memory consumption of 3D models, our model paves the way to bridge 2D and 3D vision tasks.

## 2 RELATED WORKS

### 2.1 2D SEMANTIC SEGMENTATION

The success of deep convolutional neural networks (Simonyan & Zisserman, 2014; He et al., 2016) for object classification led researchers to exploit the possibility of working out dense prediction problems. Fully Convolutional Networks (FCN)(Long et al., 2015) based encoder-decoder architectures have dominated the research of semantic segmentation since 2015. Later works extend FCN on different facets. Follow-up approaches (Badrinarayanan et al., 2017; Lin et al., 2017; Pohlen et al., 2017; Ronneberger et al., 2015; Zhao et al., 2017) leverage multi-scale feature aggregation. (Fu et al., 2019; Yin et al., 2020; Yu et al., 2020; Yuan et al., 2018; Zhao et al., 2018) apply attention mechanism to model long-range dependencies. Recently, ConNeXt(Liu et al., 2022b) modernizes the CNNs to achieve higher performance than transformers, suggesting the effectiveness of convolutional methods. On the other hand, ViT(Dosovitskiy et al., 2020) successfully introduced Transformer(Vaswani et al., 2017) to computer vision. Later, SETR(Zheng et al., 2021) demonstrates the feasibility of using Transformer-based semantic segmentation. PVT(Pyramid Vision Transformer)(Wang et al., 2021) combines pyramid structures and ViT for dense prediction. SegFormer(Xie et al., 2021) proposes a hierarchical Transformer encoder and a lightweight MLP decoder to fuse multi-level features and predict the semantic segmentation mask. Using vanilla ViT and DeiT(Touvron et al., 2021) as backbone, Segmenter(Strudel et al., 2021) designed a trainable

mask decoder to decode a patch-level embedding and interpolate to the original resolution. Combine with UPerNet(Xiao et al., 2018) structure, Swin Transformer (Liu et al., 2021) shows excellent performance on semantic segmentation.

## 2.2 3D SEMANTIC SEGMENTATION

PointNet and PointNet++ (Qi et al., 2017a;b) are the pioneer works using MLP (Multi-layer Perceptron) to directly process 3D coordinates for classification and semantic segmentation. Later works specially design convolution operations to extract features inside the neighborhood for the point clouds. Kpconv(Thomas et al., 2019) designs both rigid and deformable kernel point convolution operators for 3D point clouds nearing a set of kernel points. Minkowski Engine (Choy et al., 2019) proposes a sparse convolution to process high-dimensional data efficiently. Occuseg(Han et al., 2020) voxelized the input and uses occupancy signal to achieve instance segmentation.

The aforementioned 3D semantic segmentation works generally use 3D point cloud as representation. Apart from point cloud, volumetric representation is another widely used 3D shape representation. 3D U-Net (Çiçek et al., 2016) extends convolutional U-Net(Ronneberger et al., 2015) to 3D biomedical data segmentation. SSCNet (Song et al., 2017a) proposed to jointly do semantic segmentation and completion on RGB-D images. Scancomplete (Dai et al., 2018) propose a coarse-to-fine structure to complete scene and predict semantics. U-net structure with 3D CNN kernels is the dominant backbone in this research field.

## 2.3 TRANSFORMERS IN COMPUTER VISION

Transformer(Vaswani et al., 2017) was proposed to model long-range dependence between words in NLP tasks. Vision Transformer (Dosovitskiy et al., 2020) firstly introduce transformer to computer vision research by embedding image into non-overlapping patches, which serve as tokens for self-attention computation. Variants of ViT such as T2T-ViT(Yuan et al., 2021) , TNT(Han et al., 2021), CrossViT (Chen et al., 2021), LocalViT (Li et al., 2021),Msg-TransformerFang et al. (2022), CvT (Wu et al., 2021), and XCiT(Ali et al., 2021) optimized vanilla ViT for further image classification performance gain. Apart from image classification, PVT (Wang et al., 2021), Swin Transformer (Liu et al., 2021), CoaT (Xu et al., 2021), LeViT(Graham et al., 2021) and Twins (Chu et al., 2021) refer to the hierarchical design of FCN(Long et al., 2015), tailor transformer model for dense prediction tasks. Because of the remarkable performance of transformers in modeling long-range dependence, we adopt a pure-transformer architecture in this work.

## 2.4 KNOWLEDGE DISTILLATION

The idea of model compression first came out in 2006 where (Buciluă et al., 2006) proposed to transfer knowledge from large model to smaller model without much performance drop. (Hinton et al., 2015) systematically summarized existing knowledge distillation ideas. They demonstrated the effectiveness of student-teacher strategy and response-based knowledge distillation. Following Hinton, many works (Ge et al., 2020; Pham et al., 2021; Li et al., 2020; Xie et al., 2020; Mirzadeh et al., 2020; Yang et al., 2019; Zhang et al., 2020; Wu & Gong, 2021; Touvron et al., 2021) further explore the technique of training response-based knowledge distillation models. Since the knowledge conveyed by output layers is limited, many researchers focus on distillation through intermediate layers. Fitnets(Romero et al., 2014) firstly introduce intermediate layer distillation by using hints from the teachers' hidden layers to guide the training of the student. the idea of feature-based distillation was carried forward by following researches (Zagoruyko & Komodakis, 2016; Tung & Mori, 2019; Guan et al., 2020; Ahn et al., 2019; Heo et al., 2019; Kim et al., 2018). Recently, BP-Net(Hu et al., 2021) utilizes a bidirectional projection module with symmetric 2D and 3D architectures to achieve mutual enhancement between 2D and 3D. Compared to our distillation design, BP-Net relies on 2D-3D paired input in both training and testing process.

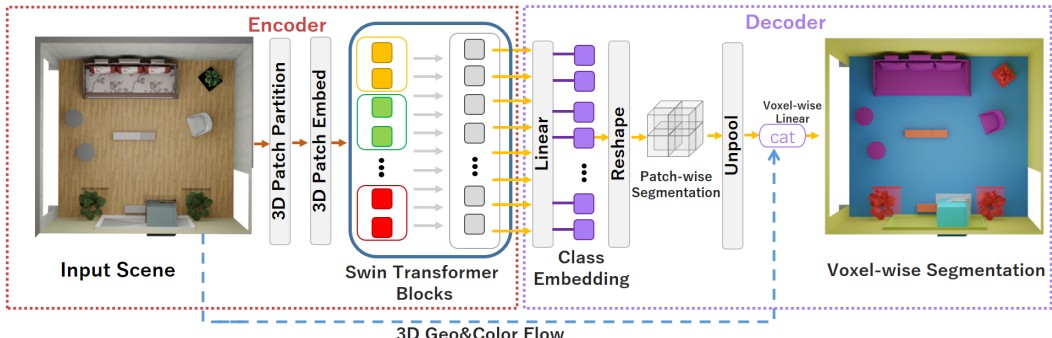

Figure 1: Overview of our 3D Segmenter pipeline. On the left-hand side is the encoder: the input blocks are partitioned and embedded into a sequence of tokens before flowing forward through 3D Swin Transformer. On the right-hand side is the Decoder: a linear-layers-only decoder with a skip connection. Receiving a sequence of tokens from the encoder, the decoder maps the class embedding to per-voxel semantic label.

## 3  3D SEGMENTER

### 3.1  ENCODER

An overview of the 3D Segmenter pipeline is shown in Figure 1.

We adapt Video Swin Transformer(Liu et al., 2022a) as our backbone. The encoder takes a Truncated Signed Distance Function (Curless & Levoy, 1996)(TSDF) block $\mathbf{x} = [x_{tsdf}, x_{rgb}] \in \mathbb{R}^{X \times Y \times Z \times 4}$ as input , where $X \times Y \times Z$ specifies the size of the block and 4 channels consist of one TSDF channel and three RGB color channels.

TSDF is an implicit regular 3D shape representation that stores distance to the nearest surface in each voxel. It is proficient in producing high-quality real-time rendering. The encoder split input into small overlappable patches before outputting a sequence of feature embedding.

Different from 2D images, when applying transformer models for 3D tasks, the length of the token sequence grows cubically with the size of the input. The shifted window design proposed by Swin Transformer(Liu et al., 2021) is efficient in limiting the sequence length of each Multi-head Self Attention (MSA) computation, which makes it has linear computational complexity to image size. For efficient and scalable 3D computation, we choose Video Swin Transformer (Liu et al., 2022a) as our encoder backbone.

Our model is trained on fixed-sized 3D blocks cropped from 3D scenes. Receiving an input block $\mathbf{x} = [x_{tsdf}, x_{rgb}] \in \mathbb{R}^{X \times Y \times Z \times 4}$, we regard each 3D patch of spatial size $P \times P \times P$ as a token. Therefore, each 3D block $\mathbf{x}$ is partitioned into $\frac{X}{P} \times \frac{Y}{P} \times \frac{Z}{P}$ tokens. A single 3D convolution layer embeds each token from 4 channels to feature dimension of size denoted by $D$.

To formulate hierarchical architecture, Video Swin Transformer reduces the number of tokens through patch merging between blocks. They use a 3D patch merging operation that concatenates the $C$ dimensional features of $2 \times 2$ spatially neighboring patches to $4C$. The token size on the temporal dimension is kept the same during merging. This reduces the number of tokens by a multiple of $2 \times 2 = 4$. To keep the resolution consistent with typical convolutional networks, the $4C$ dimensional concatenated feature is downsampled to $2C$ using a linear layer. Following this design, we use a patch merging layer to merge neighboring $2 \times 2 \times 2$ patches into $8C$ and downsample it to $2C$.

Our encoder is comprised of $h$ Swin Transformer blocks. Patch merging is performed between every two blocks. Therefore, the output of encoder is a sequence with $N = \frac{X}{P \times 2^{h-1}} \times \frac{Y}{P \times 2^{h-1}} \times \frac{Z}{P \times 2^{h-1}}$ tokens. The dimension of each token is $D \times 2^{h-1}$.

## 3.2 DECODER

The decoder maps tokens coming from the encoder to voxel-wise prediction $\hat{y}_{sem} \in \mathbb{R}^{X \times Y \times Z \times K}$, where $K$ is the number of categories in this task. In this work, we propose a decoder with only two linear layers.

Accepting token sequence of shape $N \times (D \times 2^{h-1})$ from the encoder, a token-wise linear layer is applied to the sequence, getting a $N \times K$ class embedding. The class embedding is then rearranged to patch-wise class embedding $\frac{X}{P \times 2^{h-1}} \times \frac{Y}{P \times 2^{h-1}} \times \frac{Z}{P \times 2^{h-1}} \times K$ and trilinearly interpolated to original resolution to get $y_f \in \mathbb{R}^{X \times Y \times Z \times K}$.

We use a skip connection here to 'remind' the model of the input $\mathbf{x}$. By adding a small number of parameters, the skip connection effectively enhances our performance. Concretely, we concatenate $\mathbf{x}$ and $y_f$ and feed them through a linear layer to predict the output $\hat{y}_{sem} \in \mathbb{R}^{X \times Y \times Z \times K} = Linear(Concatenate(y_f, \mathbf{x}))$ The model is trained using cross entropy loss $\mathcal{L}_{CE}$ between $\hat{y}_{sem}$ and ground truth $y_{sem}$.

## 4 2D TO 3D DISTILLATION

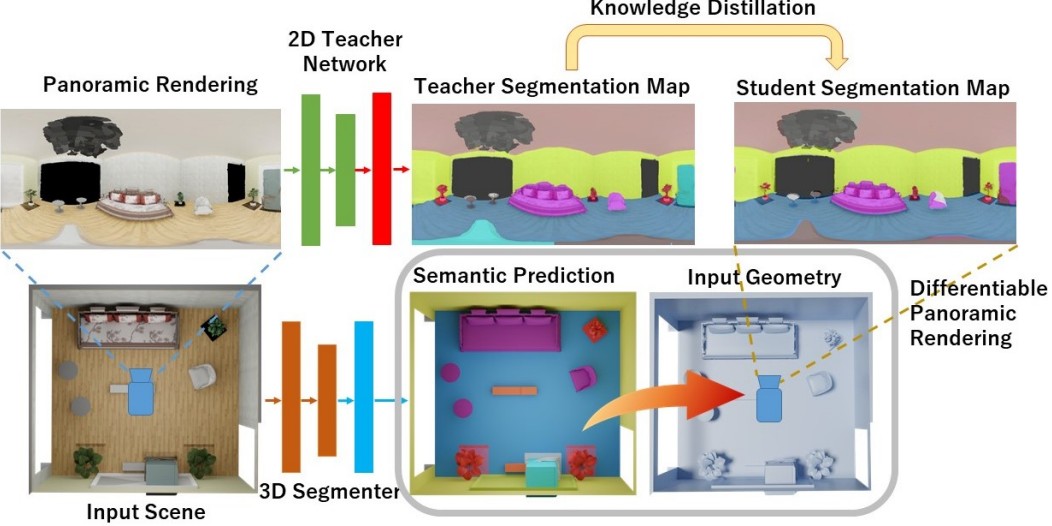

Figure 2: Overview of our proposed 2D-to-3D panoramic knowledge distillation. The panorama and 3D volume are fed into the 2D Segmentor and 3D Segmentor in parallel, and predict the semantic distributions in 2D and 3D respectively. Then we apply differentiable rendering on the predicted 3D semantic volume to obtain its panoramic projection. Finally, the knowledge is distilled from 2D to 3D projection using a Kullback–Leibler divergence loss.

As shown in Figure 2, We use pairs of panoramic renderings $I_r$ and 3D rooms $\mathbf{r} = [r_{tsdf}, r_{rgb}] \in \mathbb{R}^{X' \times Y' \times Z' \times 4}$ as our input for 2D-to-3D distillation.

The 2D teacher is pretrained on large scale 2D repository and finetuned on 2D panoramas $I_r$ and 2D semantic ground truth rendered from 3D rooms. The 3D Segmenter, as the student model in distillation, is first trained to converge using fix-sized block data. Then, the knowledge distillation from 2D teacher to 3D student is performed on the scene level.

Suppose we have a input room $\mathbf{r} = [r_{tsdf}, r_{rgb}] \in \mathbb{R}^{X' \times Y' \times Z' \times 4}$, we feed it to the student model and get voxel-level semantic segmentation $\hat{y}_r \in \mathbb{R}^{X' \times Y' \times Z' \times K}$. The 2D panoramic image could be acquired from panoramic camera placed at any position that is not occupied by objects. Here, we place it at the center of the room for sake of simplification. Combining 3D semantic segmentation prediction $\hat{y}_r$ and input geometry $r_{tsdf}$, a mapping between pixels and voxels is required to project $\hat{y}_r$ to 2D. The mapping is determined by raycasting. For each pixel in the image, we construct a ray

from the view and march along the ray through $r_{tsdf}$. Trilinear interpolation is used to determine TSDF values along the ray. The surface voxel is located when zero-crossing is detected. In this way, we establish a mapping between a pixel and a voxel. Through this mapping, 6 squared semantic segmentation maps are rendered from $+X, -X, +Y, -Y, +Z, -Z$ directions with $90°$ field of view. The 6 views are then stitched into a panorama through pixel mapping and interpolation. The total process is differentiable, making it possible for gradient backpropagates from 2D to 3D. We project a panoramic view of $\hat{y}_r$.

$$StudentSegMap = PanoramicProjecting(r_{tsdf}, \hat{y}_r) \tag{1}$$

where $StudentSegMap \in \mathbb{R}^{H \times W \times K}$, $H$ and $W$ is the height and width of rendered panorama.

After forwarding $I_r$ through the teacher model, we receive a $TeacherSegMap \in \mathbb{R}^{H \times W \times K}$ which contains the teacher model's learned knowledge. We follow (Hinton et al., 2015) to formulate a Kullback–Leibler divergence loss to measure the difference of distribution between $StudentSegMap$ and $TeacherSegMap$. We have

$$\mathcal{L}_{Dist} = \tau^2 KLDiv(StudentSegMap/\tau, TeacherSegMap/\tau) \tag{2}$$

Where $\tau$ refers to distillation temperature. The model is trained using the sum of 2D-to-3D distillation loss $\mathcal{L}_{Dist}$ and 3D cross-entropy loss $\mathcal{L}_{CE}$.

## 5 PANOROOMS3D DATASET

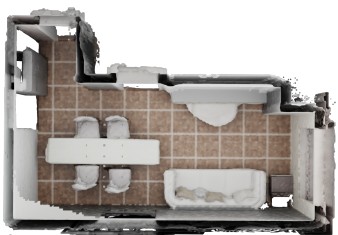 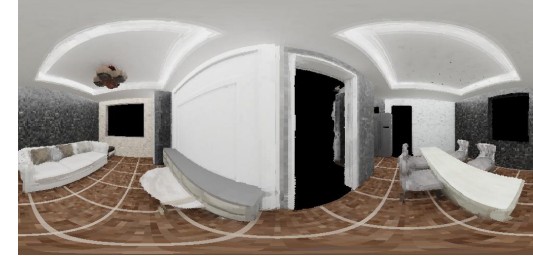

Figure 3: Sample of PanoRooms3D: a pair of input scene and its panoramic rendering.

To facilitate the knowledge distillation from the advanced 2D segmentation models to 3D models, we need a large-scale dataset of 3D scans with their corresponding panorama images. Existing datasets such as ScanNet (Dai et al., 2017) and Matterport3D (Chang et al., 2017) usually require intensive annotation effort, contain noisy labels, and do not cover complete scene geometry, while SUNCG (Song et al., 2017b) does have clean annotation but is limited to narrow field-of-view RGBD images. To this end, we propose a novel large TSDF-based semantic segmentation dataset called PanoRooms3D.

To achieve a human-like understanding of the 3D environment, we use professional designed room layouts from 3D-FRONT (Fu et al., 2021a), furnished by 3D-FUTURE(Fu et al., 2021b), a large-scale 3D CAD shapes with high-resolution informative textures.

To diversify the generation, we augment each scene with randomly sampled floor/wall textures. To obtain the TSDF representation together with semantic labels, we randomly sample camera poses inside the rooms with a height between 1.4m and 1.8m. This process simulates users holding RGB-D cameras to capture indoor scenes like (Chang et al., 2017; Dai et al., 2017; Hua et al., 2016; Armeni et al., 2017). RGB-D plus semantic frames are rendered at all camera poses and later fused into 3D scenes and semantic ground truth using TSDF fusion. A total of 2,509,873 RGB-D images and corresponding pixel-wise semantic segmentation maps are rendered. We set each voxel a $4cm$ cube. We use 120 NVIDIA Tesla V100 GPU to render for 48 hours. 3D-FRONT includes 99 semantic categories. In practice, considering the semantical consistency between labels, we consolidate similar labels, merging the original 99 classes into 30 hyper classes. Since many scenes contain furniture that is left unlabeled, we manually filter out noisy rooms. 5917 TSDF-based living rooms and bedrooms are created. We split 5325 rooms for training, 297 for validation, and 295 for

| Input Scene | 3D U-net | 3DSeg-L | Ground Truth |
|---|---|---|---|

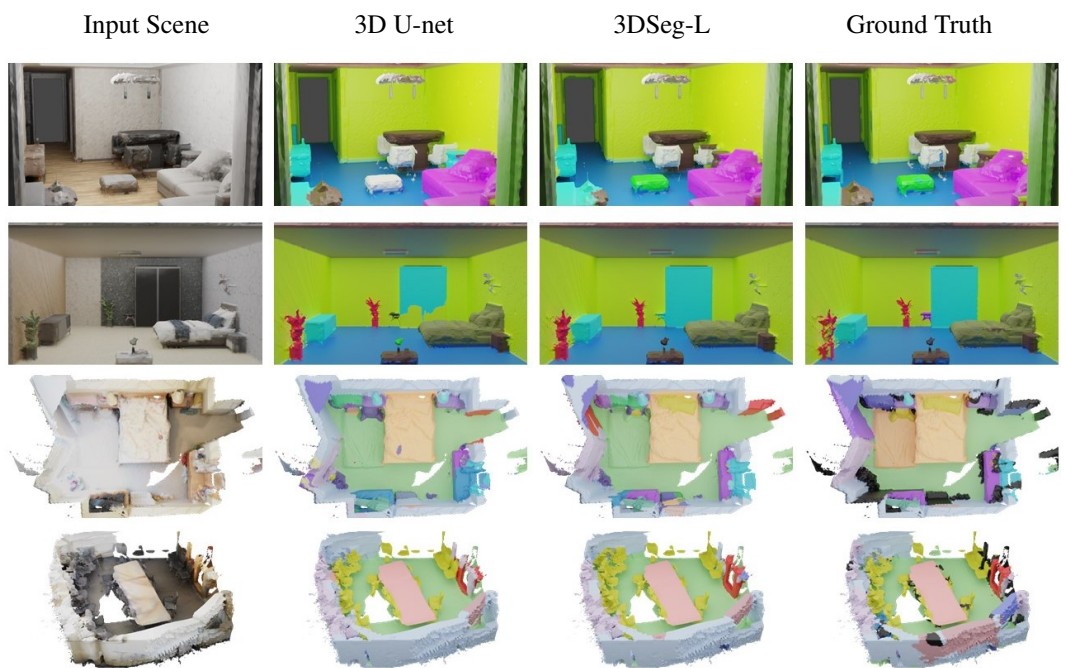

Figure 4: Qualitative comparison between our variant 3DSeg-L and convolutional baseline 3D U-Net(Çiçek et al., 2016) on our PanoRooms3D dataset(top two rows) and Scannet(Dai et al., 2017)(bottom two rows)

testing. For the convenience of window partition, we pad the all spatial dimensions of scenes to the multiples of 128.

To get panoramic images for the training of 2D teachers, we place a camera at the center of each room. The camera renders 6 square images from $+X$, $-X$, $+Y$, $-Y$, $+Z$, $-Z$ directions with a field of view of $90°$, which constitute a skybox. The skybox is then converted to an equirectangular panorama through pixel mapping and bilinear interpolation. A sample of PanoRooms3D is visualized in Figure 3.

## 6 EXPERIMENTS

### 6.1 IMPLEMENTATION DETAIL

In our training of 3D Segmenters on our PanoRooms3D dataset, we slice the 5325 rooms for training into $128 \times 128 \times 128$ cubes as the training set. The patch size $P$ is set to 4, resulting $\frac{128}{4} \times \frac{128}{4} \times \frac{128}{4} = 32768$ tokens at the beginning. The number of Swin Transformer blocks $h$ is set to 2, with $h - 1 = 1$ patch merging layer in between. The decoder receives $N = \frac{32768}{(2^{h-1})^3} = 4096$ tokens from encoder. The number of class $K$ is 30. We change the number of layers and number of MSA heads within each Swin Transformer block and embedding dimension $D$ to create different sizes of variants. Variants of encoder architecture include:

- 3DSeg-T(iny), $D = 48$, Swin Layers $[2, 2]$, Num Heads $[3, 3]$, #PARAM 0.38M
- 3DSeg-B(ase), $D = 48$, Swin Layers $[2, 4]$, Num Heads $[3, 3]$, #PARAM 0.62M
- 3DSeg-M(iddle), $D = 64$, Swin Layers $[3, 6]$, Num Heads $[4, 4]$, #PARAM 1.55M
- 3DSeg-L(arge), $D = 96$, Swin Layers $[4, 8]$, Num Heads $[6, 6]$, #PARAM 4.43M

We train our 3D Segmenter on NVIDIA Ampere A100 80GB, using the Adam optimizer (Kingma & Ba, 2014) with a learning rate of 0.001 and batch size of 8. It takes 48 hours for our models to converge.

For 2D to 3D knowledge distillation, we operate the experiments taking room as a unit. we render the 3D rooms for train/val/test sets to get 2D panoramas and 2D ground truth segmentation for train/val/test sets respectively. The 2D training set is used to finetune the 2D teacher model. The size of rendered panorama is set to $(H, W) = (512, 1024)$. We manually filter out noisy renderings in $5323$ training rooms caused by objects that block the view. We use the $5175$ panoramas after filtration paired up with their original 3D rooms to do distillation.

We choose 2D Segmenter(Strudel et al., 2021) Seg-B-Mask/16 and Seg-L-Mask/16 as the 2D teacher models. Pretrained on ImageNet-21K(Steiner et al., 2021), the models are finetuned on our rendered 2D panoramas training set for 200 epochs.

Because the scale of rooms varies, we train/validate the distillation with a batch size of 1. We train the distillation for $45000$ iterations and validate every $1500$ iterations. The distillation temperature $\tau$ is set to 20 in all of our experiments.

Apart from our PanoRoom3D dataset, we also include the Scannet (Dai et al., 2017) for baseline comparison. We random split the 1513 rooms into train/val/test 1361/77/75 respectively. All rooms are fused with a voxel resolution of $4cm$.All spatial dimensions of scenes are padded to the multiples of 128. For training, the rooms are sliced into $128 \times 128 \times 128$ cubes.

## 6.2 BASELINE COMPARISON

| | aAcc↑ | IoU↑ | Acc↑ | Dice↑ | Fscore↑ | Precision↑ | Recall↑ | #Param↓ |
|---|---|---|---|---|---|---|---|---|
| 3D U-Net | **98.98** | 58.54 | **76.88** | 65.73 | 80.89 | 65.20 | **76.88** | 16.32M |
| Sparseconv | 98.50 | 47.33 | 65.27 | 54.52 | 72.00 | 63.13 | 65.27 | 0.64M |
| VT-UNet | 98.74 | 47.90 | 71.87 | 54.75 | 75.58 | 59.46 | 71.87 | 20.89M |
| 3DSeg-T | 98.83 | 56.69 | 73.60 | 64.34 | 78.38 | 70.56 | 73.60 | 0.38M |
| 3DSeg-T Dist | 98.82 | 58.06 | 73.65 | 65.89 | 78.22 | 72.03 | 73.65 | 0.38M |
| 3DSeg-B | 98.88 | 60.41 | 75.81 | 68.28 | 80.05 | 73.62 | 75.81 | 0.62M |
| 3DSeg-B Dist | 98.87 | 61.58 | 75.21 | 69.69 | 80.18 | 75.88 | 75.21 | 0.62M |
| 3DSeg-M | 98.93 | 62.57 | 76.11 | 70.55 | 80.54 | 76.47 | 76.11 | 1.55M |
| 3DSeg-M Dist | 98.92 | 63.14 | 75.94 | 71.20 | 80.41 | 77.57 | 75.94 | 1.55M |
| 3DSeg-L | 98.95 | 64.23 | 76.48 | 72.32 | 80.98 | 78.43 | 76.48 | 4.43M |
| 3DSeg-L Dist | 98.95 | **64.67** | 76.83 | **72.84** | **81.08** | **78.86** | 76.83 | 4.43M |

Table 1: Quantitative comparison of our proposed 3D Segmenter with FCN baseline 3D U-Net(Çiçek et al., 2016), SparseConvNet(Graham et al., 2018a) and VT-UNet(Peiris et al., 2022) on our PanoRoom3D dataset. All of the metrics are reported in percentage.

| | aAcc↑ | IoU↑ | Acc↑ | Dice↑ | Fscore↑ | Precision↑ | Recall↑ | #Param↓ |
|---|---|---|---|---|---|---|---|---|
| 3D U-Net | **77.73** | 33.71 | **63.64** | 40.01 | **67.82** | 44.33 | **63.64** | 16.32M |
| 3DSeg-T | 72.33 | 28.96 | 52.22 | 36.04 | 59.54 | 44.98 | 52.22 | 0.38M |
| 3DSeg-B | 74.37 | 32.83 | 55.80 | 40.30 | 62.34 | 48.27 | 55.80 | 0.62M |
| 3DSeg-M | 76.91 | 37.65 | 59.39 | 45.33 | 65.72 | 53.13 | 59.39 | 1.55M |
| 3DSeg-L | 77.45 | **39.83** | 61.07 | **47.42** | 66.83 | **54.47** | 61.07 | 4.44M |

Table 2: Quantitative comparison of our proposed 3D Segmenter with FCN baseline 3D U-Net(Çiçek et al., 2016) on Scannet(Dai et al., 2017) dataset. All of the metrics are reported in percentage.

All of the quantitative metrics are run on a room-level. Metrics evaluated from different rooms are weighted with the corresponding rooms' volume. Here we compare our four variants and four distilled models with Fully Convolutional Network 3D U-Net(Çiçek et al., 2016), sparse convolution(Graham et al., 2018a) based model and recent transformer-based model VT-UNet(Peiris et al., 2022) . We Report 7 metrics used in semantic segmentation: overall accuracy(aAcc), mean intersection over union(mIoU), mean accuracy(Acc), Dice, Fscore, Precision, Recall. Among these metrics, mIoU is the most representative one. 3DSeg-T Dist distills knowledge from 2D Segmenter (Strudel et al., 2021) Seg-B-Mask/16. 3DSeg-B Dist, 3DSeg-M Dist and 3D Seg-L Dist distill knowledge from Seg-L-Mask/16. See the quantitative comparison in table 1. 3DSeg-B already outperforms 3D

U-Net even with only 3.8% parameters. The distillation schemes achieve a higher mIoU score without adding extra parameters. This implies that our 2D-to-3D panoramic distillation demonstrates the feasibility of training powerful 3D networks with 2D knowledge. Our baseline comparison on Scannet (Dai et al., 2017) is shown in table 2. The result shows that our proposed 3D Segmenter also achieves the best performance on real world data.

We also conduct a complexity comparison of our four baseline variants and 3D U-Net. We randomly choose 20 scenes to calculate the total inference time. Since some of the scenes are too large to be run on GPU for 3D U-Net, we run our inference time comparison on CPU. The total inference time of our four variants is $43.62s$, $44.79s$, $61.08s$, $95.89s$ from Tiny to Large, and $254.75s$ for 3D U-Net.

## 6.3 ABLATION STUDY

**The effectiveness of 2D-to-3D distillation.**    Our 3D Segmenter is pretrained on fixed-size blocks and then distilled at room level. To control variables and validate the effectiveness of our proposed distillation strategy, we conduct two experiments by finetuning 3DSeg-T and 3DSeg-B at room level with 3D cross-entropy loss only.

|                   | aAcc↑ | IoU↑  | Acc↑  | Dice↑ | Fscore↑ | Precision↑ | Recall↑ | #Param↓ |
|-------------------|-------|-------|-------|-------|---------|------------|---------|---------|
| 3DSeg-T           | 98.83 | 56.69 | 73.60 | 64.34 | 78.38   | 70.56      | 73.60   | 0.38M   |
| 3DSeg-T Finetune  | **98.85** | 56.62 | **74.58** | 64.17 | **78.72** | 69.31  | **74.58** | 0.38M   |
| 3DSeg-T Dist      | 98.82 | **58.06** | 73.65 | **65.89** | 78.22 | **72.03** | 73.65  | 0.38M   |
| 3DSeg-B           | 98.88 | 60.41 | 75.81 | 68.28 | 80.05   | 73.62      | 75.81   | 0.62M   |
| 3DSeg-B Finetune  | **98.90** | 60.87 | **76.15** | 68.79 | 80.09  | 73.95      | **76.15** | 0.62M   |
| 3DSeg-B Dist      | 98.87 | **61.58** | 75.21 | **69.69** | **80.18** | **75.88** | 75.21 | 0.62M   |

Table 3: Quantitative ablation study of distillation effectiveness. All of the metrics are reported in percentage.

We can see from Table 3 that compared with directly finetune the model on rooms, our panoramic distillation design can steadily improve the mIoU performance.

**The necessity of skip connection.**    While introducing the decoder of 3D Segmenter, we use a skip connection to 'remind' our model of the input **x**. We validate the necessity of the design in this experiment.

|                   | aAcc↑ | IoU↑  | Acc↑  | Dice↑ | Fscore↑ | Precision↑ | Recall↑ | #Param↓ |
|-------------------|-------|-------|-------|-------|---------|------------|---------|---------|
| 3DSeg-T           | 98.83 | 56.69 | 73.60 | 64.34 | 78.38   | 70.56      | 73.60   | 0.38M   |
| 3DSeg-T w/o skip  | 97.79 | 51.35 | 68.81 | 60.35 | 73.0    | 66.06      | 68.81   | 0.37M   |
| 3DSeg-B           | 98.88 | 60.41 | 75.81 | 68.28 | 80.05   | 73.62      | 75.81   | 0.62M   |
| 3DSeg-B w/o skip  | 97.86 | 54.44 | 69.18 | 63.81 | 74.12   | 70.10      | 69.18   | 0.62M   |
| 3DSeg-M           | 98.93 | 62.57 | 76.11 | 70.55 | 80.54   | 76.47      | 76.11   | 1.55M   |
| 3DSeg-M w/o skip  | 97.92 | 56.20 | 70.72 | 65.66 | 74.87   | 71.23      | 70.72   | 1.55M   |

Table 4: Quantitative ablation study of the skip connection. All of the metrics are reported in percentage.

We can see in Table 4 by adding a skip connection before output significantly improves the overall performance by adding a small number of parameters.

## 7 DISCUSSIONS

In this work, we propose the first 2D-to-3D knowledge distillation method to utilize the data abundant and pretrain-ready 2D semantic segmentation to improve 3D semantic segmentation. Experiments demonstrate our technique significantly improves the 3D model over the baseline. We have also introduced a PanoRooms3D dataset that contains a large variety of indoor scenes with dense 3D volumes and their corresponding panorama renderings.

ACKNOWLEDGMENTS

This work is supported by JST Moonshot R&D Grant Number JPMJMS2011 and JST ACT-X Grant Number JPMJAX190D, Japan and partially supported by the Shanghai Committee of Science and Technology (Grant No. 21DZ1100100)

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

# A  APPENDIX

**Limitations.** However, a few limitations are yet to be addressed. First, though the panorama images could provide the global context of the environment, the single-camera rendering inevitably suffers from occlusion, a promising direction is to directly leverage the full 2D textures from the surface of the 3D scans, as done in TextureNet (Huang et al., 2019). Second, currently, we do not consider the similarity between different models, we leave this as future work.

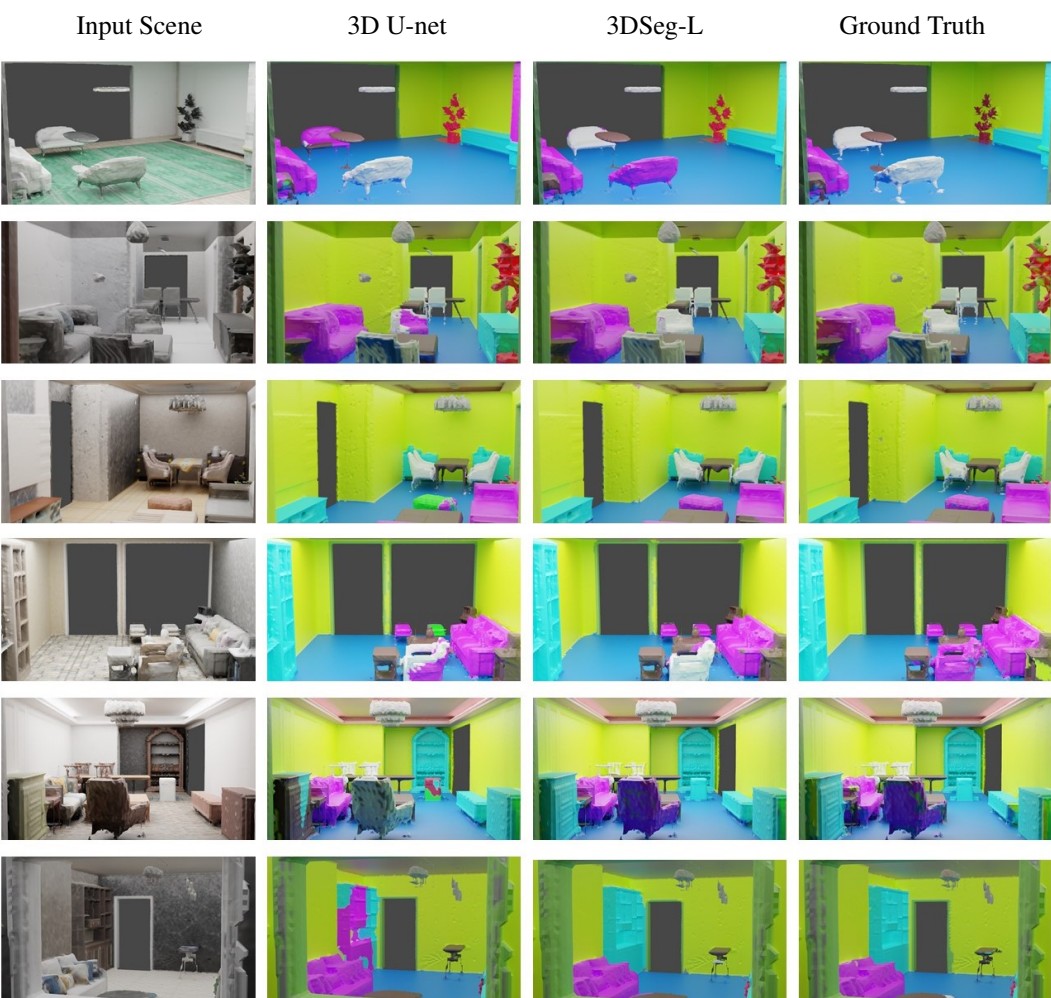

Figure 5: Qualitative comparison between our variant 3DSeg-L and convolutional baseline 3D U-Net(Çiçek et al., 2016) on our PanoRooms3D dataset.

