# OpenReview forum: "3D Segmenter: 3D Transformer based Semantic Segmentation via 2D Panoramic Distillation"
_ICLR.cc/2023/Conference — ICLR 2023 poster_

### Official Review · Reviewer_TA4Q · 2022-10-24

**Confidence:** 5
**Correctness:** 3
**Technical Novelty And Significance:** 2
**Empirical Novelty And Significance:** 2
**Recommendation:** 5

**Clarity, Quality, Novelty And Reproducibility:**

The novelty is not clear, as the author fails to make a comparison with other benchmarks. So the superiority of the method is not clear. The writing of the paper is good and easy to follow. If the author provides the dataset, I think the experiment is easy to reproduce.

**Strength And Weaknesses:**

The task of aligning 2D and 3D is important, and the current benchmarks are not good enough (including both diversity and quantity) to achieve this. The paper is good to read and easy to follow. But there miss some important experiments and comparisons:
(1) About the dataset. The ScanNet also contains about 1500 scans. Instead of providing Pano images, they have a sequence of images and corresponding labels that are used to rebuild the scan. The author should claim the strength of the proposed dataset. For example, why using pano images can achieve higher accuracy than a sequence of single images.
(2) The distillation idea to find 2D-3D corresponding samples is not new. BPNet, which is designed to combine 2D and 3D info, can also modified to the knowledge distillion.
(3) More comparisons should be provided, as current method applies sparse convolution, instead of fully 3D CNN to address the task.
(4) Some technique is not clearly stated. How to transfer a p X p X p voxel to a token embedding. Is this achieved by fully 3D CNN? If so, will this waste a lot of memory and computation on invalid space?
(5) The backbone is designed to use 3D SwinTransformer. Is the parameter initialized with pretrained video swin-transformer model? Or just training from scratch. I am curious about the effect if the params are initialized with a model from a video pretrain.

**Summary Of The Paper:**

Multi-modality between 2D and 3D is important. The paper mainly introduces a method that achieves 2D-to-3D knowledge distillation. They also propose PanoRooms3D, a large-scale 3D volumetric dataset that has aligned 2D (pano images) and 3D data (3D voxels).

**Summary Of The Review:**

See the weakness part.

---

> ### Author Response · Authors · 2022-11-08
> **Could you share detailed name of BPNet**
>
> Before submission, we have carefully surveyed existing research. By our knowledge, we are the first research proposing this similar idea. Thanks for sharing the name of BPNet but we could not find the corresponding literature. Could you provide more detailed information of this paper?

---

> ### Author Response · Authors · 2022-11-18
> **Reply to Reviewer TA4Q**
>
> >About the dataset. The ScanNet also contains about 1500 scans. Instead of providing Pano images, they have a sequence of images and corresponding labels that are used to rebuild the scan.
>
> We added an experiment on Scannet dataset. Refer to [Table 2 of this answer](https://openreview.net/forum?id=4dZeBJ83oxk&noteId=DFkQ1VUU1dV)
>
> >For example, why using pano images can achieve higher accuracy than a sequence of single images.
>
> Compared with a perspective image, a panorama image has a much larger receptive field, able to cover multiple objects in a single image. For example, in Figure 3, we visualize a sample of our dataset. On the right-hand side, the sofa, dining table, chairs, ceiling, and floor are covered in a single image.
>
> Using a sequence of images belongs to the multi-view setting, which requires knowledge fusion from multiple 2D images. We leave it as future work.
>
> >More comparisons should be provided, as current method applies sparse convolution, instead of fully 3D CNN to address the task
>
> Thanks for the suggestions on the sparse convolution works, we add a quantitative comparison with sparse convolution in Table 1 on page 8. Our models get better results than the sparse convolution-based baseline. [Jump to Table 1](https://openreview.net/forum?id=4dZeBJ83oxk&noteId=DFkQ1VUU1dV)
>
> >Some technique is not clearly stated. How to transfer a p X p X p voxel to a token embedding. Is this achieved by fully 3D CNN? If so, will this waste a lot of memory and computation on invalid space?
>
> Only a single 3D convolution layer is used to project the input patches into feature vectors. For 3DSeg-L, this projection contains 24.67 k parameters, which is around 0.557% of the total parameters. We clarified this point in Section 3.1 on page 4, marked in red.
>
> >The backbone is designed to use 3D SwinTransformer. Is the parameter initialized with pretrained video swin-transformer model? Or just training from scratch. I am curious about the effect if the params are initialized with a model from a video pretrain.
>
> We train from scratch for all of our variants in this paper. Video Swin Transformer does not apply pooling on the temporal axis. On the contrary, we pool on all axes during patch merging, as in our case, all 3 axes are physical. So we did not use pretrain on videos.

---

### Official Review · Reviewer_Vm5M · 2022-10-24

**Confidence:** 4
**Correctness:** 4
**Technical Novelty And Significance:** 2
**Empirical Novelty And Significance:** Not applicable
**Recommendation:** 6

**Clarity, Quality, Novelty And Reproducibility:**

The organization and demonstration of this paper is clear and easy to follow. The 2D-to-3D distillation idea is novel. The authors have stated that the code and dataset will be released after the acceptance of the paper.



**Strength And Weaknesses:**

Strength:
This paper is the first one to utilize the data-abundant and pretrain-ready 2D semantic segmentation to improve 3D semantic segmentation, which is reasonable and novel.
Experiments demonstrate the effectiveness of the proposed model over the 3D-Unet segmentor with convolutional architecture, and the proposed model consumes less parameters. The proposed distillation strategy is also proved to be useful.
The proposed PANOROOMS3D dataset is beneficial to the community.

Weakness:
How to get the detail correspondences between 2D panorama image pixels and 3D patches? There is lack of descriptions.
It seems that the proposed model only compares with 3D-Unet segmentor which is proposed in 2016. How about other baseline models like more powerful 3D segmentation methods proposed in recent years?
Can the proposed distilling strategy also improve 3DSeg-M and 3DSeg-L performance? There is also lack of experiments and demonstration.

**Summary Of The Paper:**

This paper proposed to solve the 3D semantic segmentation problem by distilling the knowledge embedded in the latent space of powerful 2D models. Unlike traditional knowledge distillation approaches, where student and teacher models take the same input，the inputs of the 2D teacher model in this paper are the panorama images stitched from 6 different views of 3D scenes. Thus, it is possible for gradient backpropagates from 2D to 3D. Video Swin Transformer is adapted as the 3D segmentor to segment the input 3D scenes, and a Kullback–Leibler divergence loss is used to measure the difference of distribution between student 3D segmentation maps and teacher 2D segmentation maps. To facilitate the experiments, this paper also proposed a large-scale 3D volumetric dataset with a clean voxel-wise semantic label and well-aligned corresponding 2D panoramic renderings with pixel-wise semantic labels. Experimental results on this dataset demonstrate the superiority of the proposed transformer based 3D segmentor over the 3D convolutional Unet segmentor, and the effectiveness of the proposed knowledge distillation strategy as well.



**Summary Of The Review:**

This paper proposed a knowledge distillation method to utilize the pretrain-ready 2D semantic segmentation to improve 3D semantic segmentation, which is novel and reasonable. Taking 2D panorama images collected from 3D input scenes as the input to the 2D segmentor is also beneficial to correlate the 2D-3D data. I think the overall idea of this paper is good. But other 3D segmentors except 3D Unet is also should be compared, and some details should be demonstrated more clearly.

---

> ### Author Response · Authors · 2022-11-18
> **Reply to Reviewer Vm5M**
>
> >How to get the detail correspondences between 2D panorama image pixels and 3D patches? There is lack of descriptions.
>
> We want to clarify that we are not directly making correspondence between 2D pixel and 3D patches. As written in Section 3.1 on page 4, the 3D patch is defined as a block of voxels with spatial size PxPxP. We are making correspondence between 2D pixels and 3D voxels.
>
> The pixel-to-voxel correspondence is determined by ray-casting. For each pixel in the image, we cast a ray from the camera center and march along the ray through the voxels. Trilinear interpolation is used to determine TSDF values along the ray. The surface is located when zero-crossing is detected.
> In this way, we establish a mapping between a pixel and a voxel. We revised the manuscript in Section 4 on page 5, marked in blue.
>
> The details of the ray-casting process can be found in [SPSG: Self-Supervised Photometric Scene Generation from RGB-D Scans.](https://github.com/angeladai/spsg)
>
> >It seems that the proposed model only compares with 3D-Unet segmentor which is proposed in 2016. How about other baseline models like more powerful 3D segmentation methods proposed in recent years?
>
> Also mentioned by other reviewers. Refer to [this answer](https://openreview.net/forum?id=4dZeBJ83oxk&noteId=DFkQ1VUU1dV)
>
> >Can the proposed distilling strategy also improve 3DSeg-M and 3DSeg-L performance? There is also lack of experiments and demonstration.
>
> Thanks for the advice. We conduct the 2D-to-3D distillation experiments for 3DSeg-M and 3DSeg-L  and add results in Table 1 on page 8. The result shows that our method also works for larger models. [Jump to Table 1](https://openreview.net/forum?id=4dZeBJ83oxk&noteId=DFkQ1VUU1dV)

---

### Official Review · Reviewer_NDh9 · 2022-10-24

**Confidence:** 4
**Correctness:** 3
**Technical Novelty And Significance:** 2
**Empirical Novelty And Significance:** Not applicable
**Recommendation:** 6

**Clarity, Quality, Novelty And Reproducibility:**

Quality and clarity:

The paper is clearly written and easy to follow. The figures are illustrative enough for manifesting the main idea. However, the experiment is lacking and therefore not convincing enough (see the weakness part above for more details).

Novelty:
The 2D-to-3D distillation method proposed in the paper is novel and interesting and could be inspiring for future works for 3D segmentation tasks.

**Strength And Weaknesses:**

Strength

+The 2D-to-3D distillation process is novel: the submission proposes to use differentiable panoramic rendering of 3D scene to bridge the 3D and 2D segmentation tasks.

+The proposed 3D segmentation backbone is based on the Video SwinTransformer, thus leading to high efficiency in terms of number of parameters. In the experiment, it has been shown that the 3D Transformer-based segmentation backbone archives better performance than 3DUNet with only ~4% number of parameters.

+The proposed synthetic dataset is photorealistic and complete. This could be useful for training and evaluating 3D segmentation backbones.

Weakness

-As mentioned in the limitation, the panorama rendering does not handle occlusion well. As a result, the occluded objects/parts when the camera is positioned in the center of the scene would be missing in the segmentation result.

-Experiment benchmark is limited: The proposed method is only tested on the proposed 3D segmentation synthetic dataset. The performance of the proposed pipeline on real-world datasets such as ScanNet and MatterPort3D is unknown.

-Compared method is limited: The proposed method is only compared against 3DUNet, which was proposed in 2016. There are lots of backbones and scene representations (such colored point cloud) on which the backbone has been performed since then (eg. PointTransformer).  As a result, it is hard to conclude that the proposed method archives state-of-the-art (SOTA) performance, even though it is efficient in terms of number of parameters.


**Summary Of The Paper:**

This paper proposes a pipeline for 3D scene-level segmentation based on an efficient 3D Transformer structure. On top of that, the authors also propose to distill information from pre-trained/finetuned 2D segmentation networks to improve the performance for 3D segmentation. The distillation is made possible by differentiable panoramic 2D rendering of the 3D scene. To train and evaluate the proposed model, a photorealistic synthetic 3D dataset is proposed. The performance of the proposed model is shown to be better than the 3DUnet structure but with much less parameters.


**Summary Of The Review:**

Although the idea of using the differentiable panoramic rendering for 2D-to-3D distillation is novel, the experiments are lacking in terms of both benchmarks and compared method. As a result, I’m not sure if the proposed backbone and the distillation method are good enough to achieve the SOTA performance. As a result, I’m on the fence (towards rejection) unless convincing feedbacks are provided.

---

> ### Author Response · Authors · 2022-11-18
> **Reply to Reviewer NDh9**
>
> >the panorama rendering does not handle occlusion well. As a result, the occluded objects/parts when the camera is positioned in the center of the scene would be missing in the segmentation result.
>
> The camera could be placed everywhere except those occupied by objects. To focus on the main idea in this paper,  we simplified the situation by fixing the camera at the center of the scene. We added clarification in section 4 on page 5, which is marked red.
>
> Occlusion is an inevitable problem in 3D scene understanding. One possible solution is to extend this work to a multi-view setting, i.e. placing multiple cameras in different locations of the room, which could mitigate the occlusion problem.
>
>
> >Experiment benchmark is limited: The proposed method is only tested on the proposed 3D segmentation synthetic dataset. The performance of the proposed pipeline on real-world datasets such as ScanNet and MatterPort3D is unknown.
>
> Also mentioned by other reviewers. Refer to [Table 2 of this answer](https://openreview.net/forum?id=4dZeBJ83oxk&noteId=DFkQ1VUU1dV)
>
> >Compared method is limited: The proposed method is only compared against 3DUNet, which was proposed in 2016.
>
> Also mentioned by other reviewers. Refer to [this answer](https://openreview.net/forum?id=4dZeBJ83oxk&noteId=DFkQ1VUU1dV)
>
> >There are lots of backbones and scene representations (such colored point cloud) on which the backbone has been performed since then (eg. PointTransformer). As a result, it is hard to conclude that the proposed method archives state-of-the-art (SOTA) performance, even though it is efficient in terms of number of parameters.
>
> Volumetric representation, e.g. (Truncated Signed Distance Function) TSDF, endows us the flexibility to get the 2D panoramic image corresponding to the 3D scene. In the current conference submission, we aim to explore the 2D-3D distillation strategy with this easy setting. We agree point cloud should be the direction of consequent research. In addition, we also conducted a quantitative comparison with the latest volumetric method VT-UNet(Peiris et al. 2022). The experiment shows that our method still gets superior performance.

---

> > ### Comment · Reviewer_NDh9 · 2022-11-22
> > **Post-rebuttal review**
> >
> > The rebuttal has addressed my concerns on lack of experiments comparing with other SOTA methods. The comparison results are convincing. In addition, the reviewer has addressed the my concern about the occlusion issues during training with panorama images. As a result, I think it is worth publishing given the novelty in 2D-to-3D distillation.

---

### Official Review · Reviewer_Pu1V · 2022-10-25

**Confidence:** 4
**Correctness:** 4
**Technical Novelty And Significance:** 4
**Empirical Novelty And Significance:** 4
**Recommendation:** 8

**Clarity, Quality, Novelty And Reproducibility:**

The quality of this work is very good and the work is very novel. Just the insight that knowledge distillation can be applied to models with different inputs is significant. The authors have agreed to release code and data upon acceptance so hopefully reproducing the results will be trivial, but the writing is clear enough that for a researcher with a background in 3d geometrical reasoning should be able to reproduce independently.

The main drawback is that there is not enough comparison to state of the art 3d semantic segmentation methods.  I would recommend looking at the Scannet benchmark challenge and at minimum finding some of the top entries with code and adding them to the papers comparisons.

**Strength And Weaknesses:**

Strengths:
* Proves that knowledge distillation is possible between two models that accept different inputs
* Uses this method to produce a high-quality model that performed as-good or better than one baseline model with a fraction of the parameters and runtime.
* Introduced a new dataset benchmark to encourage further development of this task.
Weaknesses:
* Only compares to one older (2016) baseline method when there are several more recent works that include code that could be used.

**Summary Of The Paper:**

This paper presents a novel method to perform knowledge distillation between a parent model that is only trained on 2d data and a child model trained on 3d data. To my knowledge this is the first paper to show that it is possible to distill knowledge between models that accept different inputs. This in itself is a novel and interesting contribution.  The model that they construct with this approach outperforms a baseline model while using only 3.8% as many parameters.

**Summary Of The Review:**

This is a novel approach for knowledge distillation between a 2d model and a 3d model for 3d semantic segmentation. The paper presents a new dataset to support this problem and presents nominal results showing that the approach is effective. Unfortunately there is not enough comparison to state of the art methods.

---

> ### Author Response · Authors · 2022-11-17
> **Reply to Reviewer Pu1V**
>
>
> >Only compares to one older (2016) baseline method when there are several more recent works that include code that could be used.
>
> Thanks to reviewers' suggestions, we compare our method with the latest work VT-UNet(Peiris et al. 2022). The result is added in Table 1 on page 8. Our method consistently outperforms VT-UNet.
>
> # Table 1
>
> |                |   aAcc↑    |  IoU↑  |  Acc↑  |   Dice↑   | Fscore↑ | Precision↑ | Recall↑ | #Param↓ |
> |:--------------:|:---------:|:-----:|:-----:|:--------:|:------:|:---------:|:------:|:------:|
> |    3D U-Net    | __98.98__ | 58.54 | __76.88__ |  65.73   | 80.89  |   65.20   |  __76.88__ | 16.32M |
> |   Sparseconv   |   98.50   | 47.33 | 65.27 |  54.52   | 72.00  |   63.13   |  65.27 | 0.64M  |
> |    VT-UNet     |   98.74   | 47.90 | 71.87 |  54.75   | 75.58  |   59.46   |  71.87 | 20.89M |
> |    3DSeg-T     |   98.83   | 56.69 | 73.60 |  64.34   | 78.38  |   70.56   |  73.60 | 0.37M  |
> |  3DSeg-T Dist  |   98.82   | 58.06 | 73.65 |  65.89   | 78.22  |   72.03   |  73.65 | 0.37M  |
> |    3DSeg-B     |   98.88   | 60.41 | 75.81 |  68.28   | 80.05  |   73.62   |  75.81 | 0.62M  |
> |  3DSeg-B Dist  |   98.87   | 61.58 | 75.21 |  69.69   | 80.18  |   75.88   |  75.21 | 0.62M  |
> |    3DSeg-M     |   98.93   | 62.57 | 76.11 |  70.55   | 80.54  |   76.47   |  76.11 | 1.55M  |
> |  3DSeg-M Dist  |   98.92   | 63.14 | 75.94 |  71.20   | 80.41  |   77.57   |  75.94 | 1.55M  |
> |    3DSeg-L     |   98.95   | 64.23 | 76.48 |  72.32   | 80.98  |   78.43   |  76.48 | 4.43M  |
> |  3DSeg-L Dist  |   98.95   | __64.67__ | 76.83 |__72.84__ | __81.08__  |   __78.86__   |  76.83 | 4.43M  |
>
> >I would recommend looking at the Scannet benchmark challenge and at minimum finding some of the top entries with code and adding them to the papers comparisons.
>
> Thanks to reviewers' suggestions, we conduct a baseline comparison on Scannet dataset. The processing of Scannet is described in Section 6.1 on page 8, marked in red, while the result of the comparison is shown in Table 2 on page 8. On this real-world dataset, we still achieve the best mean IoU performance. We also provide a qualitative comparison in the bottom two rows of Figure 4 on page 7.
>
> # Table 2
>
> |          | aAcc↑  | IoU↑  | Acc↑  | Dice↑    |Fscore↑ |Precision↑ | Recall↑ | #Param↓ |
> |:--------:|:------:|:-----:|:-----:|:--------:|:------:|:---------:|:-------:|:-------:|
> | 3D U-Net | __77.73__  | 33.71 | __63.64__ |  40.01   | __67.82__  |   44.33   |  __63.64__  | 16.32M  |
> | 3DSeg-T  | 72.33  | 28.96 | 52.22 |  36.04   | 59.54  |   44.98   |  52.22  |  0.38M  |
> | 3DSeg-B  | 74.37  | 32.83 | 55.80 |  40.30   | 62.34  |   48.27   |  55.80  |  0.62M  |
> | 3DSeg-M  | 76.91  | 37.65 | 59.39 |  45.33   | 65.72  |   53.13   |  59.39  |  1.55M  |
> | 3DSeg-L  | 77.45  | __39.83__ | 61.07 |  __47.42__   | 66.83  |   __54.47__   |  61.07  |  4.44M  |

---

### Author Response · Authors · 2022-12-11
**General Reply**

We want to thank all reviewers for their detailed and constructive feedback.

We are glad that reviewers agree that our paper is "clearly written" [NDh9, Vm5M] and "easy to follow" [NDh9, Vm5M, TA4Q].  The panoramic distillation design is "novel" [Pu1V, NDh9, Vm5M].  The work is "the first paper"[Pu1V] on distillation between different input, "bridge"[NDh9] the 2D 3D tasks, "first one to utilize the data-abundant and pretrain-ready 2D semantic segmentation to improve 3D semantic segmentation" [Vm5M]. The proposed PanoRooms3D dataset is "beneficial"[Vm5M], "encourage further development" [Pu1V], and "useful" [NDh9]. The proposed 3D Segmenter reaches "high efficiency" [NDh9]. Our dataset and code will be released upon acceptance.

We also received some constructive amendments from reviewers. Reviewers suggest for comparison with more recent methods [Pu1V,NDh9,Vm5M,TA4Q] and on other benchmark [Pu1V,NDh9,TA4Q]. We conduct further experiments with recent works on the Scannet benchmark, which is a large-scale real-world benchmark. Our method gets superior performance. We present those experiments in our rebuttal submission. Following Reviewers' [NDh9, Vm5M] suggestion, we clarify some ambiguous descriptions.

Thank you from the authors

---

### Decision · Program_Chairs · 2023-01-20

**Decision:**

Accept: poster

**Justification For Why Not Higher Score:**

There is a lack of comparison with more recent 3D segmentation methods and some missing discussion of recent work in 2D-to-3D distillation.


**Justification For Why Not Lower Score:**

The paper make several contributions that works well, including the proposed 3D segmenter, the 2D-3D distillation method, and a dataset.  The direction of 2D-to-3D distillation can be an interesting direction for the community.


**Metareview: Summary, Strengths And Weaknesses:**

Summary: The paper proposes a 2D-to-3D knowledge distillation method to improve 3D semantic segmentation.    The work proposes a 3D segmenter based on Video Swin Transformer.  For 2D-to-3D knowledge distillation, a 2D segmenter (the teacher model) is pretrained on ImageNet-21K and fined-fined on 2D panoramas, and used to train a 3D detector by using differentiable rendering of semantic volume predictions to project to panoramas.  To investigate the setting, the authors also contribute PanoRooms3D, a dataset of 5917 rooms (based on 3D-FRONT) with semantic voxels and 2D panoramic renderings.  Comparisons with prior work (3D UNet), experiments (on  PanoRooms3D and ScanNet) show that the proposed 3D Segmenter can achieve similar performance with less parameters, and that the 2D-to-3D distillation approach gives similar results as direct supervised training of the 3D detector.

Strengths:
- Proposes a 3D segmenter based on Video Swin Transformer that predicts voxel-based semantic segmentation
- The proposed 3D segmenter can achieve similar performance as prior work (3D UNet) with less parameters
- The proposed 2D-to-3D distillation approach gives similar performance as directly supervised training of the 3D detector
- Introduces a dataset with semantic voxels and aligned panoramic images

Weaknesses:
- There is limited discussion of prior work on knowledge distillation from 2D to 3D (such as [1] and [2])
- There is limited comparisons against prior work on 3D segmentation
- In the experiments, panoramic images for the pretraining and distillation are taken from the same set of rooms.  This means that during distillation, the teacher model should have relatively good predictions as the rendered panoramas would be from the same rooms it would be trained on.  A better experimental setup would be to apply the distillation on a separate set of room for which there is no annotated 3D labels.
- The number of rooms in the val/test set is small, and there are no training variances reported
- There isn't much detail about how the evaluation metrics are computed and averaged (are they macro-averaged over classes?)

[1] Image-to-Lidar Self-Supervised Distillation for Autonomous Driving Data, Sautier et al. CVPR 2022
[2] Learning from 2D: Contrastive Pixel-to-Point Knowledge Transfer for 3D Pretraining, Liu et al, 2021

Minor improvements to writing including to having space before '(' and improved typesetting of equations is recommended.


**Note From Pc:**

if the above contains the word "oral" or "spotlight" please see: "oral" presentation means -> notable-top-5% and "spotlight" means -> notable-top-25%. As stated in our emails, we are disassociating presentation type from AC recommendations